# Heart Rate Sharing at the Workplace

**Valtteri Wikström** [1,*], **Mari Falcon** [1], **Silja Martikainen** [1], **Jana Pejoska** [1,2], **Eva Durall** [2,3], **Merja Bauters** [2,4] **and Katri Saarikivi** [1]

1    Cognitive Brain Research Unit, Faculty of Medicine and Faculty of Educational Sciences, University of Helsinki, 00100 Helsinki, Finland; mari.falcon@helsinki.fi (M.F.); silja.martikainen@helsinki.fi (S.M.); jana.pejoska@helsinki.fi (J.P.); katri.saarikivi@helsinki.fi (K.S.)
2    Learning Environments Research Group, Aalto University, 02150 Espoo, Finland; eva.durall@aalto.fi (E.D.); bauters@tlu.ee (M.B.)
3    INTERACT Research Unit, Faculty of Information Technology and Electrical Engineering, University of Oulu, 90570 Oulu, Finland
4    School of Digital Technologies, Tallinn University, 10120 Tallinn, Estonia
*    Correspondence: valtteri.wikstrom@helsinki.fi

**Abstract:** Augmenting online interpersonal communication with biosignals, often in the form of heart rate sharing, has shown promise in increasing affiliation, feelings of closeness, and intimacy. Increasing empathetic awareness in the professional domain and in the customer interface could benefit both customer and employee satisfaction, but heart rate sharing in this context needs to consider issues around physiological monitoring of employees, appropriate level of intimacy, as well as the productivity outlook. In this study, we explore heart rate sharing at the workplace and study its effects on task performance. Altogether, 124 participants completed a collaborative visual guidance task using a chat box with heart rate visualization. Participants' feedback about heart rate sharing reveal themes such as a stronger sense of human contact and increased self-reflection, but also raise concerns around unnecessity, intimacy, privacy and negative interpretations. Live heart rate was always measured, but to investigate the effect of heart rate sharing on task performance, half of the customers were told that they were seeing a recording, and half were told that they were seeing the advisor's live heart beat. We found a negative link between awareness and task performance. We also found that higher ratings of usefulness of the heart rate visualization were associated with increased feelings of closeness. These results reveal that intimacy and privacy issues are particularly important for heart rate sharing in professional contexts, that preference modulates the effects of heart rate sharing on social closeness, and that heart rate sharing may have a negative effect on performance.

**Keywords:** heart rate sharing; empathy; performance; chat; instant messaging; customer service; collaborative task; computer-mediated communication

## 1. Introduction

The heart is not only an organ, but also a cultural symbol and metaphor for emotions, explored by artists and scientists throughout history [1]. More recently, transmitting physiological data has been proposed as a potential additional modality for augmenting computer-mediated communication (CMC) [2]. The underlying assumption is that availability of unintentional communication cues in CMC would support social perceptiveness and improve understanding between individuals. Increased availability of real-time physiological monitoring devices has seen a rise in these kinds of explorations, and especially heart rate sharing has been a popular topic in previous research [3–11]. As it is an emerging technology, there is still limited knowledge about heart rate sharing within real working environments, among strangers and during online text-based interaction, but at the same time, it is already practically feasible in certain real-life situations due to recent advancements in the availability of affordable and lightweight monitoring devices.

In this study, we had the opportunity to implement and test a heart rate sharing prototype with customer service personnel, whose primary work is to interact with customers over text chat. This allowed us to design the application taking into account the sensitivities of employees, whose physiological data would be shared with volunteering, actual customers in the experiment. To this end, we employed a human-centered design approach to create a system that would be acceptable to the users. The experiment itself was conducted in a meeting-room-turned-lab at the office of the company, with employees participating at their place of work and customers from their homes. Live heart rate was always transmitted, but only half the customers were told that the signal was live (referred to as "aware" customers) and the other half were told that they were seeing a recording (referred to as "non-aware" participants). To take into account the performance outlook, we created a visual guidance task with a scoring system, inspired by the customer service situation, to be completed over text chat.

Our study adds to existing literature by investigating real-time heart rate sharing in an actual work environment, with real customers and customer service advisors. Using a combination of qualitative and quantitative methods, we present findings with several consequences for designers of future applications. The results include employees experiences about heart rate sharing at the work place, and effects of heart rate sharing awareness and preference on feelings of closeness and task performance. We employ a novel study design, including an abstract pair performance task, which is modeled after a customer support situation. The visual guidance task is released as open source software (https://github.com/coblok/visual_guidance_task; accessed on 29 September 2021) for other researchers to use in their experiments.

## 2. Background

Social information processing theory predicts that CMC leads users to craft idealized self-representations, and relationships take longer to develop due to fewer emotion and intent conveying cues being available compared to in-person interaction [12,13]. The lack of non-verbal information has indeed led to expression of emotion in text-based communication to be more frequent and more explicit than in face-to-face interaction [14]. However, emotional content of email messages is typically more negatively perceived by the receiver than intended by the sender [15,16]. This is especially true when a low amount of explicit emotional cues, such as emoticons, are used [16]. Kruger et al. [15] suggest that this effect is due to the writer of the message imagining cues, such as a tone of voice indicating sarcasm or compassion, while writing the message, which is of course lost to the reader if no supporting information is available. An existing and widely adopted solution for supporting emotion contagion and empathy in text-based interaction is to increase the expressive options for including emotion-related information in messages. Emojis, sticker images, and short videos have emerged in messaging applications precisely for this purpose. However, as the choice to include them is made consciously, these solutions do not capture the range of implicit information that physical presence affords. There is also a lot of variance in the interpretation of emojis due to differences in operating systems as well as between users [17,18].

Media richness theory predicts that communication media with more cues will be better for solving equivocal tasks [19], but matching media richness to task equivocality in CMC and video communication does not seem to improve performance [20]. Furthermore, users' choice of media richness seems to be more motivated by the expected emotional reaction than the efficiency and problem-solving focused outlook, with expected pleasant outcomes leading to choice of richer media, and expected unpleasant outcomes prompting the use of less rich media [21]. Studies comparing text and voice chatting have found that text chatting during a visual task interferes with task performance more than voice chatting [22], that liking and trust between individuals in gameplay increases with the addition of audio [23], while using text chat versus audio in language learning decreases anxiety levels [24]. In comparison with face-to-face interaction, text chatting seems to

produce better results in idea generation tasks, and is characterized by increased equality of participation and less social pressure [25]. One study found that while the quality of collaborative task outcomes was similar between different media, text chats were characterized by decision making taking longer in comparison to voice chat and face-to-face conditions [26]. Text chat may have specific advantages and disadvantages in online problem solving, depending on the problem itself, the context, and on the aims of the interaction and it is important to keep in mind that text chat can also lose some of its original advantages when new information channels are added.

The customer interface is a potential application area for empathy-increasing technology, obviously to increase the satisfaction of customers, but also to humanize the customer service personnel. Customer bullying of service workers is a significant issue in the service industry [27,28], and the construction of interactions as one-off encounters tends to promote customer incivility [28]. Research in the call centre environment has identified limited cue capacity and perceived anonymity compared to face-to-face encounters to be a trigger of customer misbehaviour [29]. Customer mistreatment affects the mood and job performance of customer service employees negatively [30]. Adding information about authentic emotions and increasing the sense of connection of interacting individuals and especially increasing the customer's attention towards the emotions felt by the customer service advisor could be a way to improve collaboration in customer support, by increasing the presence and underlining the personhood of the service advisor.

Customer service text chats are a real-life situation, in which collaborative problem solving occurs over text, typically between strangers. Due to even less emotional cues being present compared to voice calls, some of the issues found in telephone support might even become exacerbated. While not yet as widely studied as call center work, customer's expressions of negative emotions have been shown to affect the performance of the service workers detrimentally also in this environment [31]. Customer service advisors' explicit textual expressions of emotion in the form of, for instance, emoticons, have been found to influence the perception of service personnel as more social, but not more reliable [32]. Instead, trust can be increased by also increasing the sense of social presence of the service person (as measured through customer ratings of a sense of human contact, personability, sociability, human warmth and human sensitivity), which also improves enjoyment and perceived usefulness of the service itself [33].

*Physio-Social Media*

Heart-rate sharing has been proposed as a way to provide implicit emotional information in CMC [3,4,6–11]. Furthermore, other types of physiological data have been explored previously, for example brain waves [34], skin conductance [5,35] and synchronization of biosignals [36]. Although physiological signals can be seen as more objective information than voluntary, and especially textual, expressions of emotion, there exists evidence that individuals can learn to control their heart rate using biofeedback [37].

The amount of information about a person's emotional state that can be derived from heart rate is relatively low. According to the circumplex model, emotions can be differentiated along two dimensions: arousal and valence [38]. Examples of high arousal emotions are exhilaration and fury, and examples of low arousal emotions are sadness and contentment. On the valence dimension, emotions range from positive to negative. In the previous example, exhilaration and contentment have positive valence and fury and sadness negative valence. Heart rate in itself does not carry valence information, and can be seen as an index of sympathetic nervous system activation, i.e., arousal [39,40]. Nevertheless, people seem to attribute emotional information to heart rate data, including information about the subject's emotional valence. One study found that in uncertain social situations, elevated heart rate was perceived to be related to negative mood [6]. Surprisingly, the same study found that normal, but not elevated, heart rate influenced trustworthiness negatively in an adversarial condition. Perceiving a partner's elevated heart rate has also been found to lead to negative mood attributions and to reduce cooperative activity in a

trust-building game, where a partner's selfish actions directly influenced the end result of the game negatively for the player [7]. Janssen et al. [3] found that during interaction in a virtual environment, hearing the sound of pre-recorded natural heart beats increased intimacy among strangers, but only when the sound was thought to originate from the partner in real-time. Based on these studies, it seems that much of the significance and many of the effects of observing the heart beat of the interlocutor arises from being conscious of the process itself and the context in which the heart rate is presented, as the actual information in the heart rate is rather limited, compared to the effects and interpretations that have been observed. This might stem from the cultural, symbolic significance of the heart and especially its relation to intimacy [1].

Physiological information can be shared either as streaming, continuous real-time data or as explicit, recorded messages. Experiments involving both have found increases in social closeness, intimacy and a sense of connection [2]. Slovák et al. [4] found that real-time heart rate visualizations can be interpreted as a source of emotional information and increase the feeling of connection between individuals. Hassib et al. [8] observed an increase in empathy among close friends and partners, who shared heart rate in an instant messaging application. This study reported the use of both real-time data and explicit messages, but found real-time data practically problematic due to users not typically using the application at the same time. Liu et al. [9] found that an application which allowed sharing recorded, explicit heart rate information in normal text messages was used as a form of emotional expression, but also as a curiosity. The same authors found that static information about a fictional drug user's heart rate increased participants emotional perspective taking towards the drug user when reading an interview, and a visual representation of the heart rate increased feelings of closeness [11]. In another study, a smart watch application was developed to explore the use of recorded heart rate messages, and the wearable nature of the device seemed to afford additional immediacy [10]. Biosignal sharing has not had only positive effects, with users being worried about negative affects unwantedly revealed by the signals, the potential for abuse, and other privacy issues [2]. Studies have also reported potential issues related to an excessive level of intimacy, especially when sharing heart rate to other people than close family and friends [2,8,9]. Users have also reported concerns about becoming distracted and mentally overloaded by the additional information received from biosignals [2].

Compared to personal communication, the professional context is characterized by productivity being highly important. Heart rate sharing should therefore support this outlook to be compatible with this environment. Interestingly, it has been found that social perceptiveness predicts group problem solving [41] and seems to be important for problem solving also during text based interaction [42]. These findings support the idea that increasing social cues during interaction could also improve collaborative problem solving. In contrast, many studies have identified that brainstorming benefits from a textual communication environment, and in general tasks that require less social-emotional communication are performed faster over text chat compared to face-to-face [25]. Additionally, at least one study has found that individual ability is more important than social perceptiveness for predicting group problem solving [43]. The effects of heart-rate sharing on task performance and problem solving remain largely unexplored. The only study we have encountered directly addressing task performance related to physio-social media found that a stress visualization derived from electrocardiogram (ECG) and skin conductance improved the performance of pairs in a worker-helper construction task over video-mediated communication [5].

It can be expected that the positive or negative performance effects of augmenting interpersonal communication with physiological cues depend to a great extent on the communication medium, the characteristics of the task at hand, on the group performing the task, and especially the way the information is presented. Depending on the application, heart rate sharing has been implemented in various ways, for example presenting the heart rate as a pulsating animation[8], as sound [3], as a graph [9], simply as a number [8,9], or

relative to a neutral level [6,7]. There are two mechanisms through which heart rate sharing can potentially have an impact on collaborative task performance. Firstly, it may provide information about the mental and physical state of the partner, improving collaboration through better understanding. Secondly, and in the case of heart rate sharing perhaps more importantly, the significance of observing another person's heart beat may increase feelings of trust and intimacy, which could improve the online customer service situation, characterized by impersonalness. Finally, it is important to point out that sharing biosignals at the workplace and between anonymous strangers has potential negative effects due to privacy and intimacy issues, based on previous studies.

## 3. Research Questions

Based on previous findings and open questions, we decided to design a prototype of a heart rate sharing text chat application, involving customer support personnel in the design process, and study this prototype with customer service advisors participating from their office and customers participating from their homes. Customer support chat provides an environment where synchronous transmission of heart rate is possible, since both interlocutors are typically present and engaged in the text chat for the duration of the discussion. In this study, we aimed to uncover some of the possibilities and problems related to heart rate sharing in a professional text chat environment and in the customer support situation, between anonymous strangers. Since the study was designed for a realistic environment, we opted for solutions feasible in the online chat-based context, focusing on the information channels and interfaces typical for that situation.

While heart rate sharing can improve collaboration in two ways, due to significance and information, as explained in the previous section, we decided to focus on the symbolic aspect. While outside-of-the-lab studies are inherently limited in their ability to provide causal interpretations, we still wanted to control for as many aspects as we could, while focusing on the symbolic meaning of the visualization. Therefore we decided to provide the same visual interface to both customer subject groups, but gave it a different meaning, with half the subjects led to believe that what they were seeing was a prototype utilizing recorded data, while the other half were correctly told that the data were live, in a similar design as in the study by Janssen et al. [3]. In this way, the visual information on the screen was of the same intensity and contained the same information. With this design, any effects we would find, especially related to social closeness and collaborative problem solving performance, would be dependent on the type of attention that was given to the visualization by the viewer, rather than on the visualization itself.

In previous heart rate sharing studies, which have reported increases in closeness or intimacy between participants [3,8,11], those factors were investigated with questionnaires, including single questions on how close the individuals felt towards each other, and interviews on the experiences of participants. In this study, we chose to measure closeness with the social closeness index adapted from Tarr et al. [44,45]. The index measures the strength of closeness, likeability, and the extent of similarity in personality between individuals, and has been previously found reliable in studying differences in closeness between pairs interacting face-to-face and in virtual reality [44,45].

We set the following research questions about sharing customer service advisors' heart rate to customers:

1.  What are the potential benefits and risks of heart rate sharing in the customer support context based on the participants' experiences?
2.  Will being aware of the real-time heart rate sharing increase the customer's feeling of social closeness with the customer service advisor?
3.  Will being aware of the real-time heart rate sharing support pair performance in the task?

Based on previous studies of heart rate sharing [3,4,8,11] we hypothesized that the customers who were aware of the real time heart rate sharing would experience more social closeness to the service advisors, although some caution is warranted as the earlier

studies are not completely comparable to the current study in their designs and user groups. Additionally, we hypothesized that such increases in social closeness could lead to better collaborative problem solving performance. Finally, thematic analysis of participants' experiences resulted in an additional research question regarding the usefulness of the heart rate visualization, which will be discussed later.

## 4. Design of the Prototype

In this section, we describe the design process of a customer support chat box interface with one-way sharing of the customer service advisor's heart rate to the customer, as well as the visual guidance task which was created for the experiment. We decided to conduct the experiment in an ecologically valid environment: The customer service advisors participated from their office, and the customers participated remotely, using their personal computers, in a typical setting for the customer service context. As we were not able to conduct heart rate measurement at the various locations from where the customers took part in the experiment, we could only measure the heart rate of the customer service advisors. This limited heart rate sharing to one direction: sharing the customer service advisor's heart rate to the customer. This design is also in line with previous results which indicate that increasing the customer's feeling of social presence of service advisors (as measured through customer ratings of a sense of human contact, personalness, sociability, human warmth and human sensitivity) can improve the customer's experience [33].

### 4.1. Human-Centered Design Approach

According to the critical theory of technology, technology is not neutral, because it can convey meanings [46]. Design choices made in communication systems may support some activities and interpretations while inhibiting others. For example, in chat-based communication, more complex avatars have been found to have a positive impact on co-presence, measured by questionnaires inquiring about the experience of being in the same place, and of belonging to the same group or community as others [47]. However, operating the interface to control those avatars can be distracting in itself [47].

Designers and scholars have argued for the need to adopt a human-centered design approach in interactive systems design in order to ensure that design solutions respond to the needs of the people who are expected to use them [48,49]. To this aim, it is considered of key importance to involve the groups who would benefit from the design at different stages of the design process. This approach goes beyond traditional ethnographic studies, which are more focused on providing rich descriptions, and opens the door to field studies that are design-oriented [50].

The adoption of participatory practices during the design process has been considered to support empathetic understanding and collaboration between designers, researchers and the design solution beneficiaries [51,52]. As a result, designers have developed myriad techniques and methods to capture people's experiences and to achieve empathetic understanding [53,54]. Sharing heart rate of the customer service advisor to the customer is a sensitive topic not only due to the implications of employers monitoring employees' physiology, but also because it involves sharing intimate information between strangers. Due to this consideration, we decided to approach the design process with a human-centered approach, taking into account the customer service advisors sentiments in the design of the system.

### 4.2. Design of a Chat Interface with Heart Rate Visualization

At an early stage, our focus became the visual communication channel, rather than audio or haptic, because we wanted the visualization to work in a typical text chat support situation for the customer, which does not necessarily include sounds or additional hardware devices. Our main design objectives consisted of visualizing the heart rate of a customer service advisor and integrating this visualization in a text chat interface. In order to achieve the design goals, we began the participatory design process with a contextual in-

quiry, in which we made observations of the existing communication and problem-solving practices between the customer service advisors and customers. We followed up with discussions about communication needs and social implications of remote communication together with the customer service advisors. After that, we continued to examine the communication tools currently in use and the practices related to them.

As part of the participatory design process, we presented our design directions to the customer service advisors so that we could iterate the designs together. At this event, three customer service advisors, in their 20s, with experience in text chat based customer support for consumer insurance, were present. The general overview presented by customer service advisors' helped us to understand how customers relate to customer service advisors when engaging in a dialogue using text chat. While initially worried about sharing their personal heart rate to customers, the customer service advisors became more positive as the discussion continued towards the possibility that seeing their heart rate would make the customer regard them more as human beings, rather than bots or representatives of a corporation.

In these discussions, the most valuable information for the design of the chat interface was related to the visual communication style. It was important to find the appropriate way to communicate the pulse in the chat interface, which could be integrated into the web pages of any service provider, while being acceptable for the customer service advisors whose heart rate would be transmitted. For these reasons, we presented to the customer service advisors design directions for representing heart rate in several visual styles: Electrocardiogram, Numeric, Symbolic, Abstract, Rhythmic and Emoji (see supplemental file).

One employee expressed her preference "I like the emoji-like heart as it is familiar", while another pointed out "I cannot maybe focus on the chat, if I see the abstract happening there". The comments and preferences about the style were informative and considerate. "The numeric can be difficult, because every person's heart rate is different, someone can have higher, someone lower. So, it maybe would not give the right information about it", commented an employee. In the end, the employees reached a consensus that a pulsating heart in emoji style, shown with a small animated heart, was preferred over the other presented graphical styles for its straight-forward interpretation and non-obtrusiveness. With this in mind, we continued to develop the visualization of the pulse to be an animated heart made in emoji style.

In addition to the choice of visualizing styles, we also discussed whether to integrate the visualization of the pulse of the customer service advisor in the chat window or behind the window. The experienced customer service advisors preferred the placement within the chat interface, to avoid possible distraction that could disturb the communication. Regarding seeing their own pulse during the interaction, the customer service advisors were of the opinion that it might distract them too much from the task and preferred not seeing their own pulse. Based on these preferences, we chose to proceed towards an imitation of the graphic design of their current chat interface. Our team created a new chat tool, with an outlook resembling the existing one, but with an additional pulsating heart symbol (see Figure 1). With this, we fulfilled the main design objectives, which consisted of a visualization of the heart rate and its integration with the chat interface.

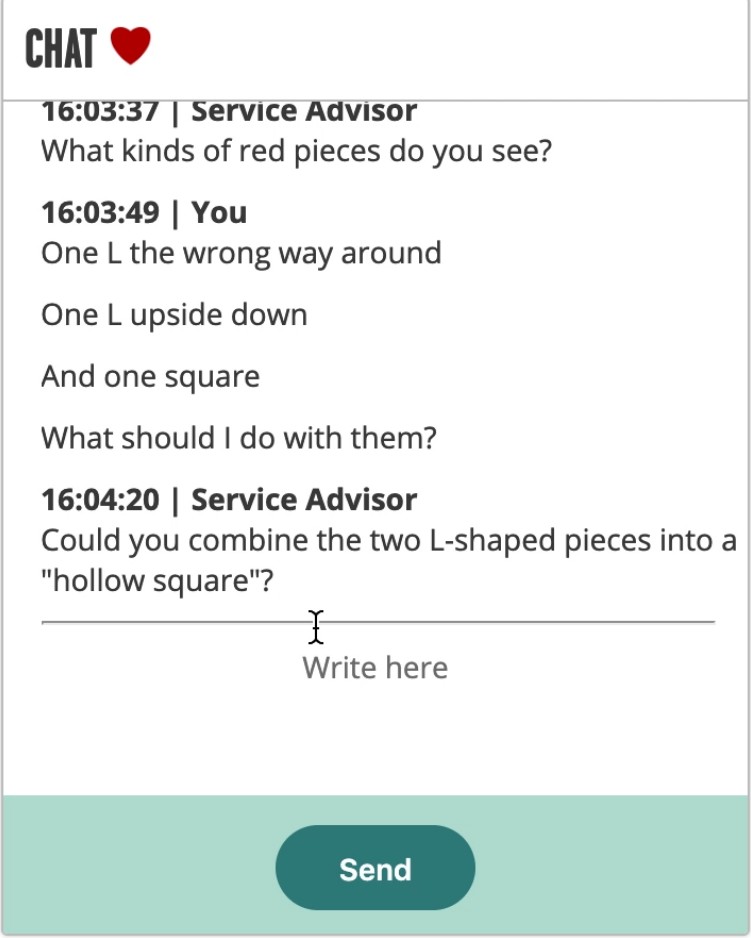

**Figure 1.** Final design of the chat interface with emoji style heart rate visualization.

*4.3. Modeling a Support Situation*

We wanted to standardize the interaction between the customer service advisor and the customer, so that we could quantify collaborative success and minimize external factors influencing the experience of test subjects. In our discussions with the customer service advisors, we uncovered that the majority of tasks they undertake are about advising customers on how to navigate the company's webpage or use the service platform (e.g., for making insurance claims and filling out other forms). Concretely, these tasks include instructing the customer in finding the right places on the service platform or webpage, clicking on the right icons etc. This entails directing the customer's visual search and motor actions, and giving feedback about customer actions. To retain these aspects of the service advisors' work, we created a visual guidance task with expert advisor and client roles, similar to the roles of "helper" and "worker" in the task created by Kraut et al. [55].

Our task consists of puzzles which are constructed on a square grid from variable shape blocks. The expert's interface consists of two grids laid next to each other, with the correct solution on the right-side grid, and the client's current state on the left-side grid. The client's interface consists of only one grid, and a set of blocks which can be placed on the grid and moved around. Each puzzle consists of a grid and a set of blocks that are all required for successful completion. In the initial state, the client sees the set of blocks next to an empty grid, and the expert sees an empty grid next to a grid which shows the correct solution. As the client places blocks on the grid and removes blocks from it, the expert sees the change of state in real-time. The client can only move the blocks, not rotate them or otherwise manipulate them. An example with client and expert views is shown in Figure 2. In this example, the client has placed one out of two blocks on the grid, in a different position than the solution requires.

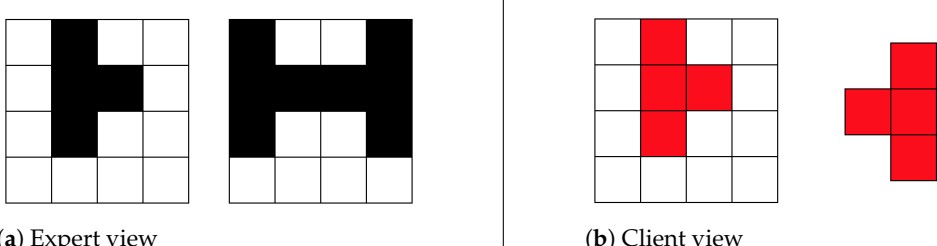

(**a**) Expert view | (**b**) Client view

**Figure 2.** A puzzle seen from the expert's (**a**) and client's (**b**) points of view. The expert sees the current state on the left-side grid and the correct solution on the right-side grid. The client sees only one grid and a set of pieces that can be placed on the grid.

A set of puzzles with varying complexity, as indicated by the amount of blocks needed to solve each puzzle, is included in the task, and they are ordered based on the amount of blocks needed to complete each puzzle (see Appendix A for puzzles included in the experiment). The puzzles are solved one at a time, and the expert indicates that a puzzle is correctly completed by clicking a button, triggering the next puzzle to be presented for both the expert and client. Participants are instructed to solve each puzzle as quickly and accurately as possible, and move on to the next puzzle as soon as they are finished with the previous one.

Performance in the task is scored based on accuracy of the results, awarding more points to solutions of more complex puzzles. In a given puzzle, the resulting grid is scored with +1 point for each correctly filled square, +0.5 points for each correct empty square and −1 point for each incorrect square, limiting the minimum total points for each puzzle to 0. If a puzzle is in progress when the time limit is reached, the score for that puzzle is calculated based on the final configuration.

## 5. The Experiment

### 5.1. Participants

The participants were employees and personal customers of an insurance company, which provides personal customers with vehicle, home, travel and health insurance. The company was selected for the study based on their widespread usage and large amount of employees working with customer service text chat. In total, 124 participants (62 pairs) were recruited via the insurance company's in-house and client mailing lists. All participants were Finnish speaking adults. Each pair comprised of a customer service advisor from the customer service team (74.2% women, mean age = 36.2 years, standard deviation (SD) = 12.1 years) and a personal customer of the company (25.8% women, mean age = 57.9 years, SD = 12.0 years). In total, 13 (21%) of the pairs were all female, 13 (21%) were all male, and 36 (58%) were mixed gender pairs. The customers and customer service advisors remained anonymous to each other throughout the experiment. Only the role of the other person was shown in the chat window as "Service Advisor" or "Customer". The University of Helsinki Ethical Review Board in the Humanities and Social and Behavioural Sciences approved the study protocol. All participants signed a written informed consent.

### 5.2. Measures

All questionnaires were filled in online. Participants answered basic demographic questions about education, work experience, income, and age. Participants' reasoning skills were assessed using an abbreviated nine-item version of the 60-item Raven's Standard Progressive Matrices Test (using items 11, 24, 28, 36, 43, 48, 49, 53, 55), which has been shown to correlate highly with the original version [56].

After the task, the participants evaluated their feeling of social closeness to their pair by answering three questions (feeling of closeness, likeability, and similarity in personality) adapted from Tarr et al. [44] using a ten-point scale. The adapted social closeness index was calculated as a mean of these three questions. We also gathered additional data, not

related to the scope of this study, on other aspects of interaction using a not yet validated questionnaire, which will be reported elsewhere.

Written open feedback data was gathered after the task with two open-ended questions, encouraging the participants to express freely what they liked and what they disliked about the heart rate visualization. We also created a questionnaire, where the participants answered seven questions on a 5-point-scale about how useful they felt that the real-time heart rate visualization (aware participants) or the recording (non-aware participants) was for interaction, conveying emotions, or supporting connection. English translations of the questions are listed in Appendix B. Seven similar questions were also posed to the service advisors about whether they felt that sharing their heart rate to the customer was useful. A sum score of the answers to all questions was calculated to indicate a total usefulness score. Internal consistency of the scale was found to be adequate among non-aware participants (Cronbach's $\alpha = 0.89$), aware participants (Cronbach's $\alpha = 0.81$) and service advisors (Cronbach's $\alpha = 0.73$).

*5.3. Procedure*

The customers participated in this study remotely, using personal computers, and were instructed by the researchers over phone and email. The customer service advisors participated from the insurance company's office where a researcher was present to instruct and conduct the physiological measurement. During the experiment, all interaction between the customer and the customer service advisor took place only within the chat environment set up specifically for the task.

After the customer had signed the consent form and committed to a time for the experiment, the background questionnaire and the reasoning skills test were sent to the customer via email. Customers were advised to fill in these questionnaires before the day of the experiment.

On the day of the experiment, written instructions for completing the task and the questionnaire links related to the experiment were sent to the customer via email approximately 30 minutes before the beginning of the experiment. The instructions were identical to all customers, with the exception of information regarding the source of the visualized heart rate. The actual heart rate of the customer service advisor was always transmitted, and all customers were told that they are testing a prototype for transmitting the service advisor's heart rate. Importantly, half of the customers were told that the data for the visualization was coming from an existing heart rate recording not related to the actual customer service advisor (referred to as "non-aware" customers) and half were told that it was a real-time visualization of the customer service advisor's actual heart rate (referred to as "aware" customers). We did not give any further instructions to the customers about how to use the heart rate information as we wanted to mimic a situation where a customer uses the prototype as a part of a customer service situation. Thus, we did not want to direct the participants expectations or use of the prototype, and wanted them to interact with the prototype in whichever way they felt appropriate.

Meanwhile, the customer service advisor was taken into one of the company's meeting rooms, which was set up with a laptop workstation used in the task and a Bitalino (r)evolution board for measuring ECG. Three ECG sensors were attached on the customer service advisor's upper body: one under each collarbone and one under the lowest right rib. The customer service advisor was then given written instructions for completing the task. The information provided to the customer service advisor was identical in all cases, regardless of the awareness condition on the customer's part. The customer service advisors participating in both conditions were told that the visualization of their real-time heart beat was presented to the customer during the experiment as "a pulsating red heart in the corner of the customer's chat window".

A few minutes before the experiment, the researcher called the customer and went through the instructions and rules for completing the task, making sure all given information regarding the task and visualization of heart rate was understood. Similarly, clarity

concerning all received information was ensured with the customer service advisor. The customer service advisor was advised to start the chat conversation in the manner of their choice and to begin the task when the customer was ready. Participants were also told that there were no restrictions when it came to communicating with each other via the chat interface during the experiment.

The participants were informed that they have 15 minutes to complete as many puzzles as possible, and that they should communicate over chat to coordinate their actions in the guidance task. The customer service advisor was instructed to move on to the next puzzle as soon as the previous one was completed. The participants were told that their task will begin once the customer service advisor has pressed the button revealing the first puzzle. Once the first puzzle had been revealed, time was observed until 15 minutes was reached. Once the time was up, the customer service advisor was asked to inform the customer that the experiment had ended and to wrap up the conversation as they wish.

Right after the experiment, both participants filled in the questionnaires for evaluating the interaction, social closeness, and open feedback about the heart rate visualization. Due to practical reasons, customer service advisors filled in the background questionnaire and the reasoning skills test after the experiment.

## 6. Thematic Analysis

The open feedback data about positive and negative aspects of the heart rate visualization were subjected to thematic analysis [57], with two researchers working together on the initial coding and deciding the final classification by reaching consensus. The individual positive and negative feedbacks were combined and isolated from other data, except for group information, which was color coded into the comments, marking comments from customer service advisors, "aware" customers and "non-aware" customers. After this, the individual comments were gathered into roughly related collections within each subject group, removing single-word answers and answers that concerned other aspects of the experiment than heart rate sharing. The comments were then rearranged into 21 smaller categories where the aware and non-aware customer groups were combined. At this stage a cross reference was done, combining positive and negative comments from one subject, if they had been grouped in the same category. These categories were combined into six overarching themes, which are analyzed below. Finally, all comments were examined for possible membership in other themes, in addition to the primary theme they had already been assigned to. In such cases the comments were duplicated and assigned to a fitting category within the additional theme. In the end, most comments were concise, and only four of the comments were assigned to two different themes.

*Results from the Thematic Analysis*

In this section, the six themes that were found through thematic analysis are presented. The description of each theme is followed by interesting and representative quotes from the study participants, more quotes can be found in Appendix C.

*Interpretation.* Many customers (10 aware and 2 non-aware customers) and customer service advisors (18) brought up topics related to interpreting and gaining information from the pulse signal. Several comments were related to receiving information about the customer service advisors mood state (7 aware customers and 6 customer service advisors). Some customer service advisors (2) and customers (1 aware, 2 non-aware) reported pulse information *calming* down the interaction. *Negative interpretation* of customer service advisors feelings, influencing the customers perception of the customer service advisor negatively was a concern noted especially among the customer service advisors (10 customer service advisors, 1 aware customer).

ʾIt was interesting to notice how the service advisor might have been excited when a new shape was presented." (Aware customer #8)

"Seeing the pulse calmed down my own doing and improved concentration." (Non-aware customer #39)

"Pulse brings out reactions that I do not want to reveal, e.g., possible irritation during the conversation." (Customer service advisor #54, this quote was also included in the Privacy theme)

*Self-reflection*. The theme of self-reflection could be identified in many comments from the customer service advisors (13). Some customer service advisors reported that knowledge of heart rate sharing had a direct effect on their mood, either as increased *excitement* (3) or through *mood regulation* (2). Additionally, a few customer service advisors (4) remarked that they would have liked to see a visualization of their *own pulse*.

"Pulse information makes me think about the way in which I communicate in chat. It's often more blunt and consisting of facts compared to, e.g., communicating on the phone." (Customer service advisor #12)

"It awakens also in myself a reaction in behavior. Keeps me maybe more patient." (Customer service advisor #30)

*Non-invasive*. Many customer service advisors (21) reported forgetting or not thinking about the heart rate transmission during the task, contrasted by only one customer service advisor remarking about electrodes being potentially disturbing. Several customer service advisors (8) reported that the task at hand had acted as a *diversion*, taking away their attention and focus from the heart rate measurement.

"I did not really pay attention or remember that the customer was seeing my pulse, because I did not see it." (Customer service advisor #42)

"To be honest I was so immersed in the chat that I did not even remember that the customer sees my pulse" (Customer service advisor #55)

*Privacy*. Several customer service advisors saw pulse information as problematic from an intimacy standpoint, considering it too personal information to share with customers (7 customer service advisors) and to strangers in general (3 customer service advisors). Customer also reported possible issues related to privacy and intimacy, bringing up issues regarding *abuse* of this kind of information (3 aware, 3 non-aware).

"In customer facing work it is not necessarily good to get too close to the customer. I want to keep my own feelings to myself, and the customer can experience my work feelings." (Customer service advisor #15)

"I feel that my own pulse is very personal information to show to a stranger" (Customer service advisor #62)

"If the pulse is live, someone might annoy on purpose to see a change in the pulse" (Non-aware customer #18)

*Presence*. The topic of presence, closeness and human touch was present in the preliminary categorization, and remained as a coherent theme throughout the analysis. In total, 9 customer service advisor and 8 customer comments (3 aware and 5 non-aware) were assigned to this category. Within this category, a sub-category of *human contact* was identified, as some customers (2 aware and 2 non-aware) specifically remarked that seeing the heart rate visualization made them feel that they were in touch with an actual human being.

"I think that transmitting heart rate information concretizes to my pair that there is a real person behind the screen. Sometimes it can feel, especially in difficult customer service situations, that the customer might forget that they are talking to a human, not to a company." (Customer service advisor #35)

"Surely, pulse information gave an understanding that I'm dealing with a human and not a robot. On the other hand, it's possible to program a pulsating heart without any real pulse measurement." (Aware customer #32)

*Unnecessity*. The pulse information was quite often thought to be irrelevant, disturbing, or not important to focus on (4 customer service advisors, 7 aware customers, 16 non-aware customers). Several of the comments were related to the pulse information being irrelevant, either generally, to the task at hand or to the customer service situation (4 customer service advisors, 5 aware customers, 6 non-aware customers). A few customers (1 aware, 3 non-aware) found the pulse visualization *distracting*. *Ignoring* or not paying attention to the

visualization was a more prevalent feedback from customers in the non-aware group (10) than in the aware group (2).

"Since it was a recording it had no connection to the real reactions of the service advisor." (Non-aware customer #31)

"In a task like this it caused issues concentrating, a little bit annoying, an unnecessity." (Aware customer #45)

"The pulse was in the background, I did not pay any attention to it." (Non-aware customer #57)

*Miscellaneous*. In addition to these six themes and the discarded comments, there were some comments from customers (7 aware) which did concern the heart visualization, but did not fit into any of the previously identified themes.

"a red heart, which pulsates is always a positive association, the rate and size does not matter as such" (Aware customer #14)

"A BEATING HEART felt safe and warm" (Aware customer #52)

### 7. Statistical Analyses

T-tests were used to compare means of participant characteristics. Multiple linear regression analyses were used to test whether being aware of the real-time heart rate sharing associated with pair performance or customers' ratings of social closeness. In the regression model, awareness of heart rate sharing was coded as 1 if aware, and 0 otherwise. We ran the analyses both unadjusted (model 1) and adjusted for the following potential covariates: age of the customer service advisor and age of the customer [58], gender (comparing mixed gender pairs to same gender pairs) [59], and pair's mean performance in the matrix reasoning task [43] (model 2).

The thematic analyses gave an indication that customer's attitudes towards heart rate sharing and the prototype were divided, with some customers reporting increased closeness and interpretations of the heart rate visualization, while others regarded the pulse information unnecessary. Based on these results, we became also interested if the perceived usefulness (see questionnaire in Appendix B) of the visualization associated with pair performance or customers' ratings of social closeness, and whether awareness of heart rating sharing moderated this association. To study this, we included an interaction term 'usefulness × awareness' into the above mentioned regression equations following the main effects. If a significant interaction was found, subanalyses were used to test whether usefulness associated differently with closeness or performance in the aware and non-aware groups. Furthermore, we tested whether the service advisors' ratings of the usefulness of the heart rate sharing was associated with their social closeness ratings to the customer or with pair performance in the task. In this case no interaction was tested as the service advisors did not know about the customer's awareness condition.

*Exclusions*

Of the 62 pairs participating in the task, one pair was excluded from the analyses for accidentally skipping the second puzzle altogether. One pair was excluded for not completing any of the puzzles after the first puzzle within the time limit due to a technical failure. Three pairs were excluded for having been allowed too little time to solve the task. Of all customers, data regarding age and/or reasoning skills was missing from five participants. No data was missing from the customer service advisors. This resulted in 52 pairs with valid data for the analyses, which were adjusted for potential covariates: 23 (44%) in the "aware" condition and 29 (56%) in the "non-aware" condition.

### 8. Results from the Statistical Analyses

*8.1. Participant Characteristics*

Pairs were able to solve 5.72 puzzles on average in 15 minutes (SD = 1.43, min = 3, max = 10). Average score from the task was 91.47 points (SD = 37.98, min = 33.50 max = 198.0 points). The customer service advisors and the customers did not differ signif-

icantly in their matrix reasoning skills (*p* = 0.90). Aware and non-aware customers did not differ in their ratings of the usefulness of the heart rate visualization (*p* = 0.48). Customer's mean usefulness rating was 19.85 points (sd = 6.35, range = 8–34 points). In total, 38.7% of the customers evaluated the visualization as useful, giving a sum score higher than 21 points on the usefulness questionnaire, which indicates that most of their answers were 3 or higher on the 5-point scale.

### 8.2. Heart Rate Sharing, Pair Performance and Social Closeness

We found that the pairs in which the customer was aware of the real-time heart rate sharing performed poorer in the task, when compared to the non-aware customers in the unadjusted regression model and also when adjusted for the potential covariates (model 1, mean difference (MD): −29.76 points, 95% confidence interval (CI): −48.47 to −11.04, *p* = 0.002, model 2, MD: −23.47 points, 95% CI: −43.94 to −3.0, *p* = 0.026). The aware and non-aware customers did not differ in their feelings of social closeness towards the service person (model 1, MD: −0.21, 95% CI: −1.18 to 0.76, *p* = 0.76, model 2, MD: 0.2, 95% CI: −1.1 to 1.4, *p* = 0.80).

We found a significant 'usefulness × awareness' interaction (model 1, *p* = 0.050, model 2, *p* = 0.031) when customers' ratings of social closeness was used as a dependent variable in the regression analyses. Subanalyses revealed that among the aware participants, customers' higher ratings of the usefulness of the visualization significantly associated with higher social closeness ratings (model 1: B = 0.19, 95% CI: 0.10 to 0.27, *p* < 0.001, model 2: B = 0.25, 95% CI: 0.10 to 0.39, *p* = 0.002), whereas no such association was found among the non-aware participants (model 1: B = 0.05, 95% CI: −0.07 to 0.17, *p* = 0.43, model 2: B = 0.06, 95% CI: −0.09 to 0.20, *p* = 0.42) (Figure 3 shows the unadjusted associations in both groups). No significant 'usefulness × awareness' interaction was found regarding pair performance (*p*-values > 0.61).

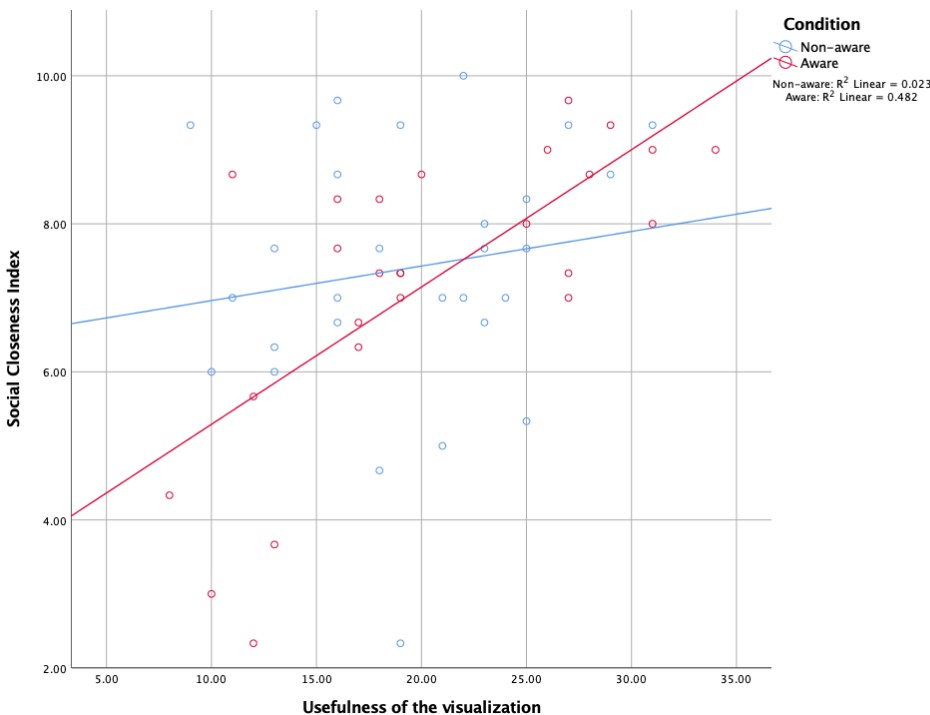

**Figure 3.** Associations between perceived usefulness of the visualization and social closeness among aware and non-aware customers.

We also found that among the service advisors, ratings of usefulness of the visualization significantly associated with their social closeness ratings towards the customer (model 1: B = 0.11, 95% CI: 0.02 to 0.20, $p$ = 0.022, model 2 B = 0.10, 95% CI: 0.01 to 0.19, $p$ = 0.028), although the association was not as strong as among the customers. No association was found between service advisors' usefulness ratings and pair performance ($p$-values > 0.47).

## 9. Discussion

We have presented the design process of a real-time heart rate sharing application for sharing customer service advisors' heart rate to customers in a text chat environment and explored the effects of the application on the service advisors' and customers' experiences, feelings of closeness and performance in an abstract text chat task. Half of the customers were aware that the visualization came from a real-time measurement and half were under the impression that it was a recording.

To our knowledge there are no previous studies assessing how introducing non-verbal psychophysiological information in a text chat environment might affect collaborative task performance. The proposition that heart rate sharing could support task performance is based on studies showing that social perceptiveness is important for collaborative problem solving face-to-face and online [41,42,60]. Providing more non-verbal information about a person's emotional state to their counterpart could increase attention towards the other. This increase in attention could also be linked to problem solving capability. In line with these hypothesis, one study has found that visualizing stress levels based on heart rate improved collaborative performance over video conferencing [5]. However, contrary to the hypothesis, our study found that pairs in which the customer was aware of the real-time heart rate sharing actually performed worse in the visual guidance task. One explanation for this finding is that the aware customers may have considered the heart rate visualization more interesting and subsequently paid more attention to the visualization, while allocating less attention to the visual guidance task, and as a result, performed worse in the task.

Physiological information can be seen as an additional and new input modality for CMC, but the output relies typically on the visual, auditory, and haptic modalities [2]. Different modalities seem to have at least somewhat separated attentional systems on the lowest levels of processing [61]. It seems that attending to simultaneous auditory and visual stimuli decreases the perceptual sensitivity to sounds only slightly, and has no effect on the visual domain, whereas simultaneous tasks presented in the same modality causes dramatic decreases in sensitivity [62]. While we decided to focus on the visual modality mainly due to practical considerations, future work should study the effect of heart rate sharing on visual task performance with an auditory or haptic presentation modality. The attentional explanation is also supported by the thematic analysis revealing a tendency of the non-aware participants to ignore the visualization. Unfortunately since the customers participated from their homes, we do not have data such as eye tracking measurements, which would provide valuable information regarding this hypothesis.

Furthermore, perceptions of elevated heart rate may negatively affect cooperation. A study by Merrill et al. [7] using the prisoner's dilemma trust game, found that when participants were led to believe that their partner's heart rate is elevated during the game, they tended to cooperate less with their partner and perceived the partner to be more anxious and less calm. In our experiment, we did not provide the customers any explicit information on whether the service person's heart rate was normal or elevated, and our task was different from a trust game, but it is possible that the participants were still prone to making negative inferences based on their counterpart's pulse, which may have interfered with collaboration.

Our results are in contrast with those obtained by Tan et al. [5], who found that a visualisation of stress level, measured by skin conductance, respiratory rate and heart rate, improved collaborative performance in a task conducted over video versus a situation with an additional video feed of the counterpart's face instead of the stress visualisation. The many differences in methodology and setup between our study and the one by Tan

et al. may explain this discrepancy in results. For instance, the difference in the amount of visual information about the counterpart could explain why performance in the stress visualisation condition was better than when video of the face was available. Actual video of the counterpart's face contains more complex, and a greater amount of dynamically changing visual information than the simple manikin that was used for stress visualisation. It is possible that in the visualisation condition, the toll of the visual stimulus on attention was smaller than in the facial video condition, facilitating task performance. This would also be in line with the explanation of our results, that attention given to the heart rate sharing visualization could have impaired task performance: The less attention allocated to information about the counterpart, the more attention can be geared towards the collaborative task at hand.

Additionally, Tan et al. [5] used more sensors, all considered sensitive to physiological arousal, which were displayed as four levels from no stress to extremely high stress. The stress visualisation gave a more explicit interpretation of the counterpart's stress level than the visualisation used in our study, which displayed the heart rate as an animated symbol growing and shrinking in visual rhythm with the heart beat, leaving far more to interpretation than stress levels. Stress as an observed emotion could, for example, provoke patience and empathy. On the other hand, the fact that we did not give the customers any indication of how to interpret the heart rate of the service advisor might have distracted the customers' attention. This would imply that the less the participants have to interpret the visualisation themselves, the more time they can allocate to the actual task. Finally, in Tan et al. [5], physiological information was shared in the opposite direction, from the worker to the helper. It is possible that the direction of information sharing is important in tasks with distinct roles. Follow-up studies investigating both the effects of visualization interpretability and the effects of the direction of physiological information sharing on task performance would be interesting and valuable.

According to the results from thematic analysis of participants' feedback, heart rate visualization increased the sense of human contact and self-reflection during chat interaction. However, participants also reported concerns about privacy and the intimate nature of the data. Additionally, customer service advisors were concerned about the possibility that customers would interpret the signals negatively, consistent with social information processing theory's prediction of users striving for idealized self-representations. Pulse information was often seen as too intimate to share to strangers, but contrary to expectations, electrodes used for measuring heart rate were not intrusive and they were often forgotten during the task. Furthermore, according to the feedback, several customers in both the aware and non-aware groups regarded heart rate sharing unnecessary or irrelevant, in contrast to others who reported interpretations and positive experiences. This led us to conduct analyses on whether perceived usefulness of the visualization was associated with feelings of closeness or task performance, and whether awareness of heart rate sharing moderated this association. Interestingly, also non-aware customers comments reflected feelings of presence and interpretations of heart rate. This could be explained by a subgroup of non-aware participants playing along with what they thought was a demonstration of a real heart rate sharing concept utilizing recorded data, as well as by the non-aware participants giving feedback on the idea of heart rate sharing, rather than on their own experiences during the experiment.

A recent systematic review of biosignal sharing by Feijt et al. [2], published after the completion of this study, points out several design consideration based on the synthesis of a large volume of work in this area. Reflecting on those design considerations, our decisions for the prototype, driven by the participatory design process and limitations of designing the experiment for a real-life environment, did not always follow the directions. First of all, our interface provided no autonomy, revisability, or reciprocity. As the participants in our participatory design workshop preferred not seeing their own heart rate, in contrast with earlier work [2], revisability was not possible, and autonomy to choose what information is shared would have contradicted with the participants worry of not being

distracted. Revisability and autonomy would have also complicated the task performance analysis. Interestingly, the customer service advisors reported a lack of distraction from the measurement itself, and even forgetting about the heart rate sharing taking place during the task, indicating that lack of autonomy and revisability did not cause significant issues while interacting with the prototype. Reciprocity was not possible due to the customer participants participating from their homes, and the measurement of heart rate still requiring specialized equipment. On the other hand, synchronicity was achieved and our visualization happened to be exactly what was suggested in the review, i.e., "a dynamic visualization of a beating heart". Sequentiality of the data transmitted continuously and simultaneously to the text messages, provided also an avenue for copresence. Finally, our system did not offer reviewability, as each heart beat was only presented once.

Previous studies of heart rate sharing have reported increases in closeness between participants [3,8,11], which was also reflected in our qualitative data. In the statistical analyses, no significant difference was found in social closeness between the aware and non-aware customer groups in general. However, among the aware customers, perception of the usefulness of the heart rate visualization was associated positively with feelings of social closeness to the customer service advisor, whereas no such association was found among the non-aware customers. Perceived usefulness was associated with social closeness also among the service advisors, who did not know that the customer participants might have been either aware or non-aware of the heart rate sharing, but knew that their own heart rate was being transmitted to the customer. This association was weaker than among the aware customers, which is understandable, as the visualization was not available to the service advisors and instead their perception of usefulness was based only on their own expectations. As such, the results partially support our hypothesis that heart rate sharing can increase social closeness, but individual appraisal regarding the usefulness of the application is identified as an important factor modulating this effect. Reflecting back on media richness theory, the variation in perceived usefulness shows the unfamiliarity of biosignal sharing. Previous experience of individuals has been shown to affect how appropriate they see new media, with more exposure leading generally to more acceptance [63]. Therefore, if the new media achieves widespread adoption, it may also be seen as more useful.

This *preference effect*, which suggests that perception of usefulness affects the quantifiable benefit, in this case feelings of social closeness, can also be linked to the symbolic significance of the heart. Although we did not observe a similar statistical difference in social closeness between the aware and non-aware groups as reported in the study by Janssen et al. [3], we did observe an interaction between awareness and perceived usefulness of the visualization, with participants in the aware group, who perceived the visualization as more useful also experiencing more closeness. Several key differences in the studies can explain this difference. First of all, in the study by Janssen et al. [3], there was no additional task, but the participant was only engaged in social observation and spatial movement with the avatar. Secondly, the use of an auditory modality to convey the heart beat did not interfere with the visual observation of the avatar, in the same way as presenting both the heart beat and the collaborative task in the visual modality in our study. It could be also, that avatars and heart beat sounds in a virtual reality environment represented such a novel situation to participants, that they were not critical to the concept of heart rate sharing itself. In our case, the interaction between customers and customer service advisors was a familiar situation for both parties, and as reflected in our thematic analysis, the addition of heart rate sharing was sometimes evaluated critically. In any case, in both our study and the study by Janssen et al., it seems that the symbolic significance of heart rate sharing is more important than the actual information, and our finding expands this to preference: the closeness effect that arises from believing to observe another person's heart rate requires that the application is considered useful and purposeful, at least in the professional context. Furthermore, the considerable variation in perceptions of

usefulness is also an explanation for the fact that no group-level difference in usefulness ratings was observed.

*Limitations and Future Work*

Based on these findings, the visualization could have a very different influence among different participants, and evidently the individual meaning that the participant gives to the visualization is an important factor in determining its benefits. As our study does not allow for inferring causal relationships, it is also possible that the participants in the aware group, who felt more social closeness, were also more prone to attribute this effect to the visualization. Additional studies that divide users into different groups based on their perceptions of usefulness and intimacy of sharing physiological data are needed to explore this effect more closely. Usefulness could also be utilized in the design of heart rate sharing interfaces, using it as a target measure in an iterative design process. Based on our analysis, additional aspects which designers should take into account are privacy and abuse concerns, intimate nature of the data, distractiveness of the visualization, and the possibility for negative interpretation of heart rate, or at least the users' concern for the possibility of negative interpretation.

The prototype was designed based on a participatory design process involving the customer service advisors, as they were seen as experienced representatives of the customer service chat environment. We based this decision also on the conclusion, that the sensitive issues related to heart rate sharing in this context were especially relevant for the customer service advisors. Having customers themselves participate in the design process would increase its participatory nature, and we think that the open-ended feedback, which was gathered from both the customers and the customer service advisors can act as a first step in this process. Future work should consider the customers feedback that we have collected, as well as include the customers in the design process from the beginning, as recipients of the heart rate information.

We conducted a study in a real-life environment, with customer service personnel participating from their office and customers from home. This can be seen as both a strength and limitation, as having real participants of varied age in a natural environment increases the ecological validity of the study, but the customer participants' actions could not be controlled as accurately as in a laboratory setting. As such, we could not monitor that the customers were following instructions, which was reflected especially as missing questionnaire data (see: Section 7).

The abstract task can also be seen as both a strength and a weakness: on one hand it allows the study session to be standardized and quantifiable, but on the other hand it does not represent a real customer service situation. Using a quantifiable task is in any case a relevant addition to the existing literature, as to our knowledge no previous studies have examined whether heart rate sharing in a text chat environment could have an effect on performance.

As the study did not include an additional condition, where a visual control object of similar activity as the beating heart would take its place, it is impossible to say what the results would have been regarding social closeness and pair performance in a situation, where there is no association to heart rate sharing whatsoever and no actual information derived from the physiology of the participants. Additionally, non-verbal information on emotions and psychophysiology can be shared using various visual, auditory, and even haptic cues, thus the scope of emotion transfer in our study is narrow and leaves plenty of room for future studies with more elaborate ways of conveying non-verbal emotional information in online text-based interaction. Going further from heart rate sharing, additional sensors and more complex analyses of the ECG, such as heart rate variability (HRV) could bring more complex and more useful information about the interlocutor. It should also be mentioned that in this study the heart rate information was only shared from the customer service advisor to the customer, thus a setting with bi-directional heart rate sharing or sharing the customer's heart rate to the customer service advisor might have different

outcomes. Since we decided to focus on a realistic scenario, we did not have the possibility of measuring and transmitting the heart rate of the customer, or to monitor additional signals, such as the eye gaze of the customers, who participated from home.

The study did not investigate whether participants were able to recognize and interpret the signal that was presented. For instance, recognition of an elevated heart rate could influence both performance in the task as well as the experience of the visualisation and the interaction. Additionally, long term usage of heart rate sharing could provide different effects: as the novelty wears off, the negative effect on task performance could disappear, and the ability to interpret the heart rate visualization could improve. Future studies which deploy heart rate sharing in repeated and continued interaction would be able to explore learning effects users may experience.

Other questions that were beyond the scope of the current study include those pertaining to how heart rate visualisation affects customer satisfaction and the quality of interaction, as well as how familiarity with the visualisation may influence the effects it has on closeness and collaboration. In addition to closeness, also other aspects of interaction may be influenced by heart rate visualisation, such as social cohesion. These effects on interaction may also contribute to customer satisfaction with the collaboration, independent of the outcomes of collaborative problem-solving. Familiarity with heart rate visualisations and concurrent processing of several visual inputs in online settings may in turn influence or mediate the effects of heart rate sharing on joint problem-solving and the quality of interaction. Taking these questions into consideration in future studies will help draw a more coherent picture on the relationship between heart rate sharing and its effects on the success of collaboration and experiences of interaction.

## 10. Conclusions

We have reported the design process of a real-time heart rate sharing text chat system intended for use in customer support and tested it with real customers and customer service advisors completing a novel quantifiable collaborative visual guidance task. Our findings indicate that visualized heart rate sharing may not be an optimal strategy for supporting online problem-solving in this context. Task performance in our experiment actually suffered compared to the control group, who were led to believe that the heart rate visualization they were seeing did not originate from their partner's physiology. This finding may be related to the heart rate visualization taking attention away from the task at hand. In general, heart rate sharing elicited a variety of different opinions and appraisals ranging from perceptions of increased closeness and self-reflection to violations of privacy. Furthermore, finding heart rate sharing to be useful was shown to be associated with an increased feeling of social closeness to one's partner. Based on our results, heart rate sharing in professional contexts should be considered carefully for a given situation, as it might interfere with actual problem solving performance, and sharing heart rate to strangers, as well as physiological monitoring of employees, raises intimacy and privacy concerns. Additionally, if heart rate sharing is used in this environment, it is recommended to be an optional feature, as effects of increased social closeness depend on the application being considered useful.

**Supplementary Materials:** The following are available online at https://www.mdpi.com/article/10.3390/mti5100060/s1, Video S1: The interface in use and a visualization of the task progression of the participants.

**Author Contributions:** Conceptualization, V.W., M.F., S.M., J.P., E.D., M.B. and K.S.; data curation, V.W., M.F., S.M. and E.D.; formal analysis, S.M.; funding acquisition, V.W. and K.S.; methodology, V.W., S.M., J.P. and E.D.; software, V.W.; supervision, M.B. and K.S.; visualization, V.W., S.M. and J.P.; writing—original draft, V.W., M.F., S.M., J.P., E.D., M.B. and K.S.; writing—review and editing, V.W. All authors have read and agreed to the published version of the manuscript.

**Funding:** This research was funded by grants from Business Finland and Ella and Georg Ehrn-rooth's foundation.

**Institutional Review Board Statement:** The study was conducted according to the guidelines of the Declaration of Helsinki, and approved by The University of Helsinki Ethical Review Board in the Humanities and Social and Behavioural Sciences (protocol code 19/2018 approved on 9 May 2018).

**Informed Consent Statement:** Informed consent was obtained from all subjects involved in the study.

**Data Availability Statement:** The data presented in this study are available on request from the corresponding author. The data are not publicly available due to protection of the employee participants identity in front of their employer, as the data contains psychometric information, and the written consent obtained from participants restricts publishing their data openly.

**Acknowledgments:** We would like to thank Kalle Ahokas, Cerina Eromäki, Aleksi Haatainen, Katri Kennedy, Miia Nurmi, Mikko Virmasalo and everyone else involved at If for being enthusiastic and helpful, as well as for the great effort of acquiring subjects, making a study with real employees and customers possible.

**Conflicts of Interest:** The authors declare no conflict of interest.

## Abbreviations

The following abbreviations are used in this manuscript:

| CMC | Computer-mediated communication |
|-----|---------------------------------|
| ECG | Electrocardiogram |
| HRV | Heart rate variability |

## Appendix A. Puzzles Included in the Study

**Table A1.** Puzzles included in the study.

| Solution | Blocks |
|---|---|

### Appendix B. Usefulness Questionnaires

**Table A2.** Questions about the usefulness of the heart rate visualization (scale 1-totally disagree to 5-totally agree).

| Questions to the Aware Participants | Questions to the Non-Aware Participants |
|---|---|
| I feel that heart rate sharing helped me understand how the service person is feeling. | I feel that seeing the heart beat recording helped me understand how the service person is feeling. |
| I feel that heart rate sharing helped me to think about how the service person is feeling. | I feel that seeing the heart beat recording helped me to think about how the service person is feeling. |
| I feel that heart rate sharing made me pay more attention to my style of conversation. | I feel that seeing the heart beat made me pay more attention to my style of conversation. |
| I felt more connected to the service person because of the heart rate sharing. | I felt more connected to the service person because of the heart beat. |
| Heart rate sharing made it more difficult to me to focus on the actual conversation with the service person. (Reverse item) | Seeing the heart beat made it more difficult to me to focus on the actual conversation with the service person. (Reverse item) |
| I feel that heart rate sharing could be useful for interaction. | I feel that seeing the heart beat recording could be useful for interaction. |
| I would like to use a similar application in a real customer service situation. | I would like to use a similar application in a real customer service situation. |

### Appendix C. Participants' Comments

In this appendix we have collected prototypical comments and comments rich in information from the open-ended feedback, organized in the categories which were formed through thematic analysis. The provided quotes have been translated from Finnish to English, while trying to preserve the original meaning and style as accurately as possible.

*Appendix C.1. Presence and Closeness*

"Pulse gives for sure the image to the person on the other end of the chat, that the human on the other end is also human and cares." (Customer service advisor #25)

"Closeness with another person" (Customer service advisor #11)

"I liked that the customer could maybe better conform to my situation and possibly think about the interaction situation also from my point of view." (Customer service advisor #4)

"It felt like I knew the buddy, like I was replying to a friend." (Non-aware customer #12)

"I think that transmitting heart rate information concretizes to my pair that there is a real person behind the screen. Sometimes it can feel, especially in difficult customer service situations, that the customer might forget that they are talking to a human, not to a company." (Customer service advisor #35)

Human Contact

"Surely, pulse information gave an understanding that I'm dealing with a human and not a robot. On the other hand, it's possible to program a pulsating heart without any real pulse measurement." (Aware customer #32)

"Pulse established that I knew I was dealing with a real person, not a robot" (Non-aware customer #55)

*Appendix C.2. Interpretation*

"low pulse predicts positivity, high pulse is a hint of negativity towards customer." (Aware customer #3)

"It was interesting to notice how the service advisor might have been excited when a new shape was presented." (Aware customer #8)

"the customer can better follow my mood state." (Customer service advisor #5)

"The customer saw my pulse and it for sure affected their thoughts while doing the task. Improves understanding." (Customer service advisor #60)

"The customer had a possiblity to perceive if I'm nervous, and if they are doing correct." (Customer service advisor #51)

Appendix C.2.1. Calming

"I felt calm all the time, so I believe this can also calm down the customer." (Customer service advisor #45)

"Seeing the pulse calmed down my own doing and improved concentration." (Non-aware customer #39)

Appendix C.2.2. Negative Interpretations

"I experienced the transmission of pulse information a little bit distressing. What if something would have gone badly in the interaction and the customer would have seen a rising pulse. A bad state of mind and stress could have caught on to the customer." (Customer service advisor #8)

"Pulse brings out reactions that I do not want to reveal, e.g., possible irritation during the conversation." (Customer service advisor #54)

"I did not like the visibility of my own frustration and failure to the customer." (Customer service advisor #48)

"Afterwards I was thinking whether the other can interpret my pulse correctly. The tasks were going great, except the last one which we did not finish began poorly. At that moment my pulse might have risen, but I do not know if the other felt that I was frustrated or agitated, although I was trying to think how to better express myself." (Customer service advisor #43)

"In a customer service situation, faceless communication does not feel as good, even if the discussion feels otherwise easy. Seeing anothers pulse can even be harmful to the interaction, if you see that the others heart is beating very fast. It can be interpreted, e.g., as insecurity and it's not necessarily a good thing." (Aware customer #39)

*Appendix C.3. Privacy*

"In customer facing work it is not necessarily good to get too close to the customer. I want to keep my own feelings to myself, and the customer can experience my work feelings." (Customer service advisor #15)

"In my opinion biometric information should not be used in this context. My heart graph or my frustration level does not belong to the customer, and I do not even consider it good customer service if the customer receives any kind of information about my emotional state. I'm a representative of a company, in which the customer is doing a transaction, I'm interested in their matters and well-being, the customer does not need to be interested in me, but the company and taking care of their matter. If heart graph or other biometrics are used in customer encounters, it should be in the way that I see the stress levels of the customer, and can adapt my service style and attitude accordingly." (Customer service advisor #44)

"I feel that my own pulse is very personal information to show to a stranger" (Customer service advisor #62)

"In a sales situation it's not good to show emotional states or pulse." (Non-aware customer #38)

Abuse

"Transmitting pulse may be too 'intimate' of a thing to transmit. If someone considers being rude to the customer servant as their life's mission, seeing their reactions might act as a trigger" (Aware customer #4)

"If the pulse is live, someone might annoy on purpose to see a change in the pulse" (Non-aware customer #18)

*Appendix C.4. Unnecessity*

"I do not think that knowing the pulse or transmitting the pulse influences in any way my interaction." (Customer service advisor #58)

"I do not think that seeing a pulse symbol has any role in this task. I did not look at it during the task, because I was focusing on the task. I do not think that seeing the customer advisor's pulse has any significance from the customers point of view. It is more important that the advisor is empathetic and able to solve the customer's problems in a satisfactory manner." (Non-aware customer #36)

"Since it was a recording it had no connection to the real reactions of the service advisor." (Non-aware customer #31)

"A real man is not influenced by heart beats" (Aware customer #9)

Appendix C.4.1. Distracting

"In a task like this it caused issues concentrating, a little bit annoying, an unnecessity." (Aware customer #45)

"A flashing heart makes the chat screen restless" (Non-aware customer #49)

Appendix C.4.2. Ignoring

"The pulse was in the background, I did not pay any attention to it." (Non-aware customer #57)

"I did not react to the recorded pulse almost at all. I was focusing on the service advisor's words and the tasks :)" (Non-aware customer #40)

"I paid almost no attention to it during the task" (Aware customer #20)

*Appendix C.5. Self-Reflection*

"Pulse information makes me think about the way in which I communicate in chat. It's often more blunt and consisting of facts compared to, e.g., communicating on the phone." (Customer service advisor #12)

"It was nice to focus on the situation in a different way, knowing that my pulse is being transmitted to the customer." (Customer service advisor #47)

Appendix C.5.1. Excitement

"Even though I liked to be conscious of my own actions because of the pulse measurement, I think it also gave me a certain level of tension. If the test was longer, it would be possible to relax more and get other results." (Customer service advisor #16)

Appendix C.5.2. Mood Regulation

"It helped to keep my own 'excitement' at bay." (Customer service advisor #56)

"It awakens also in myself a reaction in behavior. Keeps me maybe more patient." (Customer service advisor #30)

Appendix C.5.3. Own Pulse

"It would have been nice to see my own pulse." (Customer service advisor #20)

"Sometimes I was thinking what my pulse looks like, was it changing, I was not at all conscious about it." (Customer service advisor #38)

*Appendix C.6. Non-Invasive*

"I actually did not even think about pulse measurement and the customer seeing it." (Customer service advisor #26)

"I did not really pay attention or remember that the customer was seeing my pulse, because I did not see it." (Customer service advisor #42)

"I do not think there is a problem with pulse transmission. Having electrodes in a real situation could potentially be cumbersome." (Customer service advisor #22)

Diversion

"I did not pay attention to the measurement. I was so focused on the task." (Customer service advisor #13)

"To be honest I was so immersed in the chat that I did not even remember that the customer sees my pulse" (Customer service advisor #55)

"I did not even remember the pulse :) The task was fun. I like the task." (Customer service advisor #28)

*Appendix C.7. Miscellaneous*

"a red heart, which pulsates is always a positive association, the rate and size does not matter as such" (Aware customer #14)

"A BEATING HEART felt safe and warm" (Aware customer #52)

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
