# Peer review of "Heart Rate Sharing at the Workplace"

_mti, doi:10.3390/mti5100060_

Round 1
Reviewer 1 Report
The authors have adequately addressed my concerns - I'm happy to accept the paper.
Author Response
Thank you for your review and helpful suggestions!
Reviewer 2 Report
The authors have addressed my main concern adequately. I think this is solid work that should be published.
Author Response
Thank you for your review and helpful suggestion!
Reviewer 3 Report
The paper reports on a study in which the effects of a visualisation of heart rate were investigated for a text-chat customer service interaction. There are two types of results: First, qualitative statements on the overall effects, usefulness and reservations; second, a quantitative comparison between two groups where one had a real-time experience, and one observed a recording (or believed so).
The general topic is interesting on a long-term and basic research perspective. The study was designed and carried out carefully and with high standards. The only potential problem in the study design is that the control group observing a recorded heart rate visualisation was put in a rather strange, unrealistic and artificial situation, which may limit external validity.
The results obtained are not spectacular but may provide helpful inout for further research. Probably the most interesting result is that task performance was negatively affected by adding the heart rate visualisation, probably due to some attention split effect. This observation may be helpful for designing applications using such a feature.
Presentation, structure and language are good, and I have found just a few very minor issues to be corrected in the text (see below).
Altogether, I vote for acceptance after the minor revisions have been carried out.
Minor text revisions:
- p 2, Section 2, "has lead to" should be "has led to"
- p 11, Section 6, after ref [58] there is a quotation mark which should be removed
Author Response
Thank you for the review and the minor text revisions. Both errors have been corrected, as well as another lead -> led correction in the Conclusion.
This manuscript is a resubmission of an earlier submission. The following is a list of the peer review reports and author responses from that submission.
Round 1
Reviewer 1 Report
This work addresses a general topic which will become ever more relevant over the coming years, which is the usage of physiological sensor information in the interaction with digital systems. The concrete use case studied is visual feedback embedded into a customer service chat application, indicating the heart rate of the service advisor to the customer.
While being sympathetic to the general goal of this research, I am very sceptical about the concrete study presented here. My main points of criticism are as follows:
(1) Technology-centered approach: Although the paper claims at some point to follow a user-centered design approach, the general question studied is of the type "How can we use a certain piece of technology for something useful" (the technology being the heart rate monitor here). This means starting from an arbitrary solution and looking for the right problem to solve. Proper user-centred design would have started from the needs and demands of the users, not from the technology. Moreover, it would have taken into account other stakeholders besides the two directly involved persons.
(2) Raw heart rate is a rather doubtful physiological indicator, since heart rate varies greatly depending on a person's individual physiology, in particular better physical fitness generally leads to lower resting heart rate. Also the sensor data indicates whether the person has just arrived (maybe over a staircase) or just is back from a brief walk to the restroom, just to give a few examples. Other stress and load indicators may make much more sense in the studied situation, for instance based on Heart Rate Variability (HRV).
(3) The measurement of the most important dependent variable, pair performance, is based on solving a geometric puzzle. It is argued that this is a model for customer service tasks, where the advisor helps the customer to navigate through Web pages. It is very strong and doubtful assumption to take this as a general model for customer service dialogues. This task completely excludes the standard situation where the client asks the service personnel to perform some task for them. It is, by the way, a very bad sign if the company sees their customer service as a riddle to be solved by the customer.
(4) The study design compared two groups of "aware" and "not-aware" customers, where the "not-aware" group was told that the heart rate data came from an arbitrary, unrelated recording. This condition does not make sense at all for the customer (it essentially only distracts them). So I do not see the point in comparing a very artificial and useless scenario with a moderately useful scenario.
(5) The study was carried out and analysed in decent quality, however the insights gained from the study are very limited. It was observed that performance of the "aware" group in the task was poorer, which can be easily explained by the cognitive load generated from matching the heart rate information with the task progress. So this is not very surprising. Another result is that in the "aware" group, higher subjective ratings of usefulness for the visualisation are associated with higher perceived social closeness. This does not tell us much, since it mainly confirms that the customers built a subjective interpretation of the heart rate as indicating or increasing closeness. This is some confirmation of previous research but in itself not a strong result.
So altogether, my impression of this paper is that is basically interesting work, but does not present journal-grade results. I would rather see this as a good work-in-progress submission to an HCI conference. Therefore, I vote for rejection of the paper. However, I would be willing to have a look at a revised version, in case the overall decision leads to the paper being resubmitted.
Reviewer 2 Report
Overall, I liked this paper. I think it's an original and well-executed study, and a well-written paper. The results are rich, adequately analysed - using both quantitative and qualitative measures to arrive at insights – and with an appropriate discussion of the hypotheses, as well as the implications and limitations of the study. Having said this, I do feel the discussion can gain somewhat in theoretical depth (see my more detailed comments below – esp. 7 and 8). I like the fact that authors have made an effort to situate the experiment in an ecologically valid setting - even though this necessarily limits causal attributions. The topic of (design to support) interpersonal psychophysiology is relevant and of interest to the readers of this journal. As we are moving to a world where interpersonal interactions are increasingly mediated - for better or for worse - these kinds of studies help scholars to consider the future of such mediation, and the boundary conditions and contexts where physio-social media may potentially add value, or where they are unlikely to do this .
I have some comments and questions that may help the authors in improving the paper.
- The first paragraph of the Background Section (lines 52-60) feels somewhat disconnected to the way the study is introduced in the first Section. While the introductions mainly mentions a focus on task performance and feelings of closeness, the Background Section suddenly starts about Customer bullying. I think the flow of the paper could be improved here (e.g., mentioning customer bullying as part a potential application area, but not as a rationale for doing this study).
- In the part on physio-social media, the authors mention the potential issues related to an excessive level of intimacy very briefly (lines 161-163), while elaborating much more on positive effects of intimacy. In my opinion, this does not accurately reflect the frequency of users' concerns regarding their privacy, fear of misinterpretation and corresponding misunderstandings that are being reported in these studies.
I'd like to point the authors to a very recent review on Biosignal sharing, that has been accepted to appear in the journal Human-Computer Interaction. The authors could not have been aware of this reference at the time of submitting their paper, but since it is so obviously related to their work, I will point to it here:
Feijt, M. A., Westerink, J. H. D. M., de Kort, Y. A. W., & IJsselsteijn, W. A. (2021). Sharing Biosignals: An Analysis of the Experiential and Communication Properties of Interpersonal Psychophysiology. Human-Computer Interaction, XX(XX).
- The study does not strictly test the effect of heart rate feedback, as there is no control condition, but only differentiates between 'aware' and 'non-aware' conditions. I would recommend making this more explicit from the start, as that changes the expectations readers have. It also makes it less surprising to me that you find no differences on social connectedness, especially since you did not investigate how participants interpreted the signals. This would have provided information that might have helped to understand why there were no differences between the two conditions on your primary measures, but why you did find an interaction effect.
- In the design process (lines 252-276), you only included customer advisors, but not customers. As the visualization of the heart rate is especially important for them, their opinion & experiences also seem relevant here. On what grounds did you make this choice? This is a limitation of the prototype that can and should be reflected upon in the discussion.
- In the section on the thematic analysis (lines 467-485), there is no description of how many researchers have coded the data and whether a coding check was performed. Please include this information to indicate the reliability of the results.
- Some of the quotes of the theme Negative interpretation (e.g., "Pulse brings out reactions that I don’t want to reveal, e.g. possible irritation during the conversation") to me feel overlapping with Privacy (e.g. ""I feel that my own pulse is very personal information to show to a stranger"). On which basis did you make a distinction between these codes and decide to which theme to assign such a quote?
- I like how the discussion connects the findings of the current study to commonalities and differences in design and outcome of related studies. One such studies is the study by Janssen et al, where I was personally involved as co-author. I’m not convinced that the authors’ interpretation on the overall novelty of the situation (lines 745-747) points to the most relevant difference between the studies. In Janssen et al., the study was about testing the potency of heart-rate sharing are a nonverbal cue, in relation to predictions drawn from equilibrium theory, that predicts a trade-off between nonverbal cues to maintain comfortable levels of interpersonal intimacy. I see two more relevant differences between the studies that the authors probably should highlight: (i) As the focus of the Janssen study was solely on the social communication/interaction (e.g., tasks included looking the avatar in the eyes or at the chin, and to move towards the avatar as closely as was comfortable), there was no additional task performance that would distract participants’ attention away from the heartrate signal. This may have allowed more attentional capacity to be dedicated towards processing the potential significance of the HR signal. (ii) Janssen et al used the auditory modality to communicate heartrate. This did not conflict with the other social cues which were visual in nature, thus allowing for their continuous and simultaneous processing. In the current study, both the task (in the chat) and the HR feedback (also in the chat) were using the visual modality for feedback and communication, which creates a conflict or trade-off in sensory processing and attentional capacity – it’s essentially a zero-sum game: one cannot attend to both at the same time. Both (i) and (ii) offer, to my mind, more logical explanations for the found differences between the studies. As the journal has a focus on Multimodal interfaces, it would make sense to highlight the relevance of differentiating between multimodal versus unimodal attentional processes, and add a few references to relevant work in this area (e.g., the Spence & Driver work).
- As an overall comment, I feel the discussion could be improved in terms of depth when not merely discussing results in a comparative fashion (i.e., in relation to other studies) but to also consider theoretical explanations and implications (e.g., of the nature I discussed under point 7, but not limited to this). The theories and references of the Introduction could offer a good starting point in this respect, and could be returned to more explicitly.
I hope these comments are helpful in improving the paper. I enjoyed reading it, and I think the paper deserves publication after these recommendations have been incorporated.
Reviewer 3 Report
This was an excellently performed study. The writing was clear, comprehensive, and careful. The results were well presented. The overall contribution is moderate because the design intervention itself was incremental and limited due to the inherent problems of the ambiguity of heart-rate data. However, it does build upon other work on heart-rate and makes worthwhile contributions. The authors point out the limitations of the work. The only suggestion I have to improve it is to rename "aware" and "non-aware" to something more accurate like "live" and "recorded" (heart-rate). Non-aware could mean many things that it doesn't (e.g., readers may erroneously assume it means that the person didn't notice the heart icon).